# Blind Calibration in Compressed Sensing using Message Passing Algorithms

**Christophe Schülke**
Univ Paris Diderot, Sorbonne Paris Cité,
ESPCI and CNRS UMR 7083
Paris 75005, France

**Francesco Caltagirone**
Institut de Physique Théorique
CEA Saclay and CNRS URA 2306
91191 Gif-sur-Yvette, France

**Florent Krzakala**
ENS and CNRS UMR 8550,
ESPCI and CNRS UMR 7083
Paris 75005, France

**Lenka Zdeborová**
Institut de Physique Théorique
CEA Saclay and CNRS URA 2306
91191 Gif-sur-Yvette, France

## Abstract

Compressed sensing (CS) is a concept that allows to acquire compressible signals with a small number of measurements. As such it is very attractive for hardware implementations. Therefore, correct calibration of the hardware is a central issue. In this paper we study the so-called blind calibration, i.e. when the training signals that are available to perform the calibration are sparse but unknown. We extend the approximate message passing (AMP) algorithm used in CS to the case of blind calibration. In the calibration-AMP, both the gains on the sensors and the elements of the signals are treated as unknowns. Our algorithm is also applicable to settings in which the sensors distort the measurements in other ways than multiplication by a gain, unlike previously suggested blind calibration algorithms based on convex relaxations. We study numerically the phase diagram of the blind calibration problem, and show that even in cases where convex relaxation is possible, our algorithm requires a smaller number of measurements and/or signals in order to perform well.

## 1   Introduction

The problem of acquiring an $N$-dimensional signal $\mathbf{x}$ through $M$ linear measurements, $\mathbf{y} = F\mathbf{x}$, arises in many contexts. The Compressed Sensing (CS) approach [1, 2] exploits the fact that, in many cases of interest, the signal is $K$-sparse (in an appropriate known basis), meaning that only $K = \rho N$ out of the $N$ components are non-zero. Compressed sensing theory shows that a $K$-sparse $N$-dimensional signal can be reconstructed from far less than $N$ linear measurements [1, 2], thus saving acquisition time, cost or increasing the resolution. In the most common setting, the linear $M \times N$ map $F$ is considered to be known.

Nowadays, the concept of compressed sensing is very attractive for hardware implementations. However, one of the main issues when building hardware revolves around calibration. Usually the sensors introduce a distortion (or decalibration) to the measurements in the form of some unknown gains. Calibration is about how to determine the transfer function between the measurements and the readings from the sensor. In some applications dealing with distributed sensors or radars for instance, the location or intrinsic parameters of the sensors are not exactly known [3, 4]. Similar distortion can be found in applications with microphone arrays [5]. The need for calibration has been emphasized in a number of other works, see e.g. [6, 7, 8]. One common way of dealing with calibration (apart from ignoring it or considering it as measurement noise) is *supervised calibration*

when some known training signals $\mathbf{x}_l$, $l = 1, \ldots, P$ and the corresponding observations $\mathbf{y}_l$ are used to estimate the distortion parameters. Given a sparse signal recovery problem, if we were not able to previously estimate the distortion parameters via supervised calibration, we will need to estimate the unknown signal and the unknown distortion parameters simultaneously - this is known as *blind (unsupervised) calibration*. If such blind calibration is computationally possible, then it might be simpler to do than the *supervised calibration* in practice. The main contribution of this paper is a computationally efficient message passing algorithm for blind calibration.

## 1.1 Setting

We state the problem of *blind calibration* in the following way. First we introduce an unknown distortion parameter (we will also use equivalently the term decalibration parameter or gain) $d_\mu$ for each of the sensors, $\mu = 1, \ldots, M$. Note that $d_\mu$ can also represent a vector of several parameters. We consider that the signal is linearly projected by a known $M \times N$ measurement matrix $F$ and only then distorted according to some known transfer function $h$. This transfer function can be probabilistic (noisy), non-linear, etc. Each sensor $\mu$ then provides the following distorted and noisy reading (measure) $y_\mu = h(z_\mu, d_\mu, w_\mu)$ where $z_\mu = \sum_{i=1}^N F_{\mu i} x_i$. As often in CS, we focus on the case where the measurement matrix $F$ is iid Gaussian with zero mean. For the measurement noise $w_\mu$, one usually considers an iid Gaussian noise with variance $\Delta$, which is added to $z_\mu$.

In order to perform the blind calibration, we need to measure several statistically diverse signals. Given a set of $N$-dimensional $K$-sparse signals $\mathbf{x}_l$ with $l = 1, \cdots, P$, for each of the signals we consider $M$ sensor readings

$$y_{\mu l} = h(z_{\mu l}, d_\mu, w_{\mu l}), \quad \text{where} \quad z_{\mu l} = \sum_{i=1}^N F_{\mu i} x_{il}, \tag{1}$$

where $d_\mu$ are the signal-independent distortion parameters, $w_{\mu l}$ is a signal-dependent measurement noise, and $h$ is an arbitrary known function of these variables with standard regularity requirements. To illustrate a situation in which one has sample dependent noise $w_{\mu l}$ and sample independent distortion $d_\mu$, consider for instance sound sensors placed in space at positions $d_\mu$ that are not exactly known. The positions, however, do not change when different sounds are recorded. The noise $w_{\mu l}$ is then the ambient noise that is different during every recording.

The final inference problem is hence as follows: Given the $M \times P$ measurements $y_{\mu l}$ and a perfect knowledge of the matrix $F$, we want to infer both the $P$ different signals $\{\mathbf{x}_1, \cdots \mathbf{x}_P\}$ and the $M$ distortion parameters $d_\mu$, $\mu = 1, \cdots M$. In this work we place ourselves in the Bayesian setting where we assume the distribution of the signal elements, $P_X$, and the distortion coefficients, $P_D$, to be known.

## 1.2 Relation to previous work

As far as we know, the problem of blind calibration was first studied in the context of compressed sensing in [9] where the distortions were considered as multiplicative, i.e. the transfer function was

$$h(z_{\mu l}, d_\mu, w_{\mu l}) = \frac{1}{d_\mu}(z_{\mu l} + w_{\mu l}). \tag{2}$$

A subsequent work [10] considers a more general case when the distortion parameters are $d_\mu = (g_\mu, \theta_\mu)$, and the transfer function $h(z_{\mu l}, d_\mu, w_{\mu l}) = e^{i\theta_\mu}(z_{\mu l} + w_{\mu l})/g_\mu$. Both [9] and [10] applied convex optimization based algorithms to the blind calibration problem and their approach seems to be limited to the above special cases of transfer functions. Our approach is able to deal with a general transfer function $h$, and moreover for the product-transfer-function (2) it outperforms the algorithm of [9].

The most commonly used algorithm for signal reconstruction in CS is the $\ell_1$ minimization of [1]. In CS without noise and for measurement matrices with iid Gaussian elements, the $\ell_1$ minimization algorithm leads to exact reconstruction as long as the measurement rate $\alpha = M/N > \alpha_{\mathrm{DT}}$ in the limit of large signal dimension, where $\alpha_{\mathrm{DT}}$ is a well known phase transition of Donoho and Tanner [11]. The blind calibration algorithm of [9, 10] also directly uses $\ell_1$ minimization for reconstruction.

In the last couple of years, the theory of CS witnessed a large progress thanks to the development of message passing algorithms based on the standard loopy Belief Propagation (BP) and their analysis [12, 13, 14, 15, 16]. In the context of compressed sensing, the canonical loopy BP is difficult to implement because its messages would be probability distributions over a continuous support. At the same time in problems such as compressed sensing, Gaussian or quadratic approximation of BP still contains the information necessary for a successful reconstruction of the signal. Such approximations of loopy BP originated in works on CDMA multiuser detection [17, 18]. In compressed sensing the Gaussian approximation of BP is known as the approximate message passing (AMP) [12, 13], and it was used to prove that with properly designed measurement matrices $F$ the signal can be reconstructed as long as the number of measurements is larger than the number of non-zero component in the signal, thus closing the gap between the Donoho-Tanner transition and the information theoretical lower bound [15, 16]. Even without particular design of the measurement matrices the AMP algorithm outperforms the $\ell_1$-minimization for a large class of signals. Importantly for the present work, [14] generalized the AMP algorithm to deal with a wider range of input and output functions. For some of those, generalizations of the $\ell_1$-minimization based approach are not convex anymore, and hence they do not have the advantage of provable computational tractability anymore.

The following two works have considered blind calibration related problems with the use of AMP-like algorithms. In [19] the authors use AMP combined with expectation maximization to calibrate gains that act on the signal components rather than on the measurement components as we consider here. In [20] the authors study the case when every element of the measurement matrix $F$ has to be calibrated, in contrast to the row-constant gains considered in this paper. The setting of [20] is much closer to the dictionary learning problem and is much more demanding, both computationally and in terms of the number of different signals necessary for successful calibration.

### 1.3 Contributions

In this work we extend the generalized approximate message passing (GAMP) algorithm of [14] to the problem of blind calibration with a general transfer function $h$, eq. (1). We denote it as the calibration-AMP or Cal-AMP algorithm. The Cal-AMP uses $P > 1$ unknown sparse signals to learn both the different signals $\mathbf{x}_l$, $l = 1, \ldots, P$, and the distortion parameters $d_\mu$, $\mu = 1, \ldots, M$, of the sensors. We hence overcome the limitations of the blind calibration algorithm presented in [9, 10] to the class of settings for which the calibration can be written as a convex optimization problem.

In the second part of this paper we analyze the performance of Cal-AMP for the product transfer function (2) used in [9] and demonstrate its scalability and better performance with respect to their $\ell_1$-based calibration approach. In the numerical study we observe a sharp phase transition generalizing the phase transition seen for AMP in compressed sensing [21]. Note that for the blind calibration problem to be solvable, we need the amount of information contained in the sensor readings, $PM$, to be at least as large as the size of the vector of distortion parameters $M$, plus the number of the non-zero components of all the signals, $KP$. Defining $\rho = K/N$ and $\alpha = M/N$, this leads to $\alpha P \geq \rho P + \alpha$. If we fix the number of signals $P$ we have a well defined line in the $(\rho, \alpha)$-plane given by

$$\alpha \geq \frac{P}{P-1}\rho \equiv \alpha_{\min} , \tag{3}$$

below which exact calibration cannot be possible. We will compare the empirically observed phase transition for blind calibration to this theoretical bound as well as to the phase transition that would have been observed in the pure CS, i.e. if we knew the distortion parameters.

## 2 The Calibration-AMP algorithm

The Cal-AMP algorithm is based on a Bayesian probabilistic formulation of the reconstruction problem. Denoting $P_X(x_{il})$ the assumed empirical distribution of the components of the signal, $P_W(w_{\mu l})$ the assumed probability distribution of the components of the noise, and $P_D(d_\mu)$ the assumed empirical distribution of the distortion parameters, the Bayes formula yields

$$P(\mathbf{x}, \mathbf{d}|\mathbf{F}, \mathbf{y}) = \frac{1}{\mathcal{Z}} \prod_{i,l=1}^{N,P} P_X(x_{il}) \prod_{\mu=1}^{M} P_D(d_\mu) \prod_{l,\mu=1}^{P,M} \int dw_{\mu l} P_W(w_{\mu l})\delta\left[y_{\mu l} - h\left(z_{\mu l}, d_\mu, w_{\mu l}\right)\right] ,$$

$$\tag{4}$$

where $\mathcal{Z}$ is a normalization constant and $z_{\mu l} = \sum_i F_{\mu i} x_{il}$. We denote the marginals of the signal components $\nu_{il}^x(x_{il}) = \int \prod_\mu \mathrm{d}d_\mu \prod_{jn \neq il} dx_{jn} P(\mathbf{x}, \mathbf{d} | \mathbf{F}, \mathbf{y})$ and those of the distortion parameters $\nu_\mu^d(d_\mu) = \int \prod_{\gamma \neq \mu} \mathrm{d}d_\gamma \prod_{il} dx_{il} P(\mathbf{x}, \mathbf{d} | \mathbf{F}, \mathbf{y})$. The estimators $x_{il}^*$ that minimizes the expected mean-squared error (MSE) of the signals and the estimator $d_\mu^*$ of the distortion parameters are the averages w.r.t. the marginal distributions, namely $x_{il}^* = \int \mathrm{d}x_{il} \, x_{il} \, \nu_{il}^x(x_{il})$ and $d_\mu^* = \int \mathrm{d}d_\mu \, d_\mu \, \nu_\mu^d(d_\mu)$. An exact computation of these estimates is not tractable in any known way so we use instead a belief-propagation based approximation that has proven to be fast and efficient in the CS problem [12, 13, 14]. We remind that GAMP, that leads to a considerably simpler inference problem, is recovered if we set $P_D(d_\mu) = \delta(d_\mu - 1)$ and that usual AMP is recovered by setting $h(z, d, w) = z + w$ on top of it.

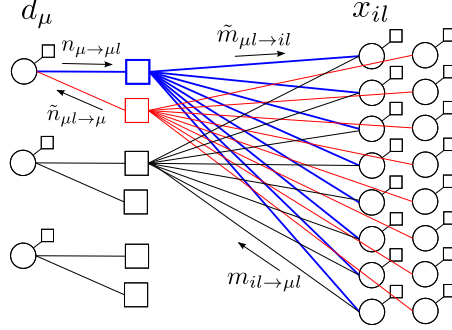

Figure 1: Graphical model representing the blind calibration problem. Here the dimensionality of the signal is $N = 8$, the number of sensors is $M = 3$, and the number of signals used for calibration $P = 2$. The variable nodes $x_{il}$ and $d_\mu$ are depicted as circles, the factor nodes as squares.

Given the factor graph representation of the calibration problem in Fig. 1, the canonical belief propagation equations for the probability measure (4) are written in terms of $NPM$ pairs of messages $\tilde{m}_{\mu l \to il}(x_{il})$ and $m_{il \to \mu l}(x_{il})$, representing probability distributions on the signal component $x_{il}$, and $PM$ pairs of messages $n_{\mu \to \mu l}(d_\mu)$ and $\tilde{n}_{\mu l \to \mu}(d_\mu)$, representing probability distributions on the distortion parameter $d_\mu$. Following the lines of [12, 13, 14, 15], with the use of the central limit theorem, a Gaussian approximation, and neglecting terms that go to zero as $N \to \infty$, the BP equations can be closed using only the means and variances of the messages $m_{il \to \mu l}$ and $n_{\mu \to \mu l}$:

$$a_{il \to \mu l} = \int \mathrm{d}x_{il} \, m_{il \to \mu l}(x_{il}) \, x_{il} \,, \qquad v_{il \to \mu l} = \int \mathrm{d}x_{il} \, m_{il \to \mu l}(x_{il}) \, x_{il}^2 - a_{il \to \mu l}^2 \,, \quad (5)$$

$$k_{\mu \to \mu l} = \int \mathrm{d}d_\mu \, n_{\mu \to \mu l}(d_\mu) \, d_\mu \,, \qquad l_{\mu \to \mu l} = \int \mathrm{d}d_\mu \, n_{\mu \to \mu l}(d_\mu) \, d_\mu^2 - k_{\mu \to \mu l}^2 \,. \quad (6)$$

Moreover, again neglecting only terms that go to zero as $N \to \infty$, we can write closed equations on quantities that correspond to the variables and factors nodes, instead of messages running between variables and factor nodes. For this we introduce $\omega_{\mu l} = \sum_i F_{\mu i} a_{il \to \mu l}$ and $V_{\mu l} = \sum_i F_{\mu i}^2 v_{il \to \mu l}$. The derivation of the Cal-AMP algorithm is similar to those in [12, 13, 14, 15]. The resulting algorithm is in the leading order equivalent to the belief propagation for the factor graph from Fig. 1. To summarize the resulting algorithm we define

$$\tilde{G}(y, d, \omega, v) = \int \mathrm{d}z \, \mathrm{d}w \, P_W(w) \, \delta[h(z, d, w) - y] \, e^{-\frac{1}{2} \frac{(z - \omega)^2}{v}} \,, \quad \text{and} \quad (7)$$

$$G(y_{\mu \cdot}, \omega_{\mu \cdot}, V_{\mu \cdot}, \theta) = \ln \left[ \int \mathrm{d}d \, P_D(d) \prod_{n=1}^P \tilde{G}(y_{\mu n}, d, \omega_{\mu n}, V_{\mu n}) \, e^{\theta d} \right] \,, \quad (8)$$

where $\mu \cdot$ indicates a dependence on all the variables labeled $\mu n$ with $n = 1, \cdots, P$, and $\delta(\cdot)$ is the Dirac delta function. Similarly as Rangan in [14], we define $P$ output functions as

$$g_{\mathrm{out}}^l(y_{\mu \cdot}, \omega_{\mu \cdot}, V_{\mu \cdot}) = \frac{\partial}{\partial \omega_{\mu l}} G(y_{\mu \cdot}, \omega_{\mu \cdot}, V_{\mu \cdot}, \theta = 0) \,. \quad (9)$$

Note that each of the output functions depend on all the $P$ different signals. We also define the following input functions

$$f_a^x(\Sigma^2, R) = [x]_X \,, \qquad f_c^x(\Sigma^2, R) = [x^2]_X - [x]_X^2 \,, \quad (10)$$

where $[\dots]_X$ indicates expectation w.r.t. the measure

$$\mathcal{M}_X(x,\Sigma^2,R) = \frac{1}{Z(\Sigma^2,R)}P_X(x)\ e^{-\frac{(x-R)^2}{2\Sigma^2}}\ . \tag{11}$$

Given the above definitions, the iterative calibration-AMP algorithm reads as follows:

$$V_{\mu l}^{t+1} = \sum_i F_{\mu i}^2 v_{il}^t\ , \qquad \omega_{\mu l}^{t+1} = \sum_i F_{\mu i} a_{il}^t - V_{\mu l}^{t+1} e_{\mu l}^{t+1}\ , \tag{12}$$

$$e_{\mu l}^{t+1} = g_{\text{out}}^l(y_{\mu\cdot},\omega_{\mu\cdot}^t,V_{\mu\cdot}^t)\ , \qquad h_{\mu l}^{t+1} = -\frac{\partial}{\partial\omega_{\mu l}}g_{\text{out}}^l(y_{\mu\cdot},\omega_{\mu\cdot}^t,V_{\mu\cdot}^{t+1})\ , \tag{13}$$

$$(\Sigma_{il}^{t+1})^2 = \left[\sum_\mu F_{\mu i}^2 h_{\mu l}^{t+1}\right]^{-1}\ , \qquad R_{il}^{t+1} = a_{il} + \left[\sum_\mu F_{\mu i} e_{\mu l}^{t+1}\right](\Sigma_{il}^{t+1})^2\ , \tag{14}$$

$$a_{il}^{t+1} = f_a^x((\Sigma_{il}^{t+1})^2,R_{il}^{t+1})\ , \qquad v_{il}^{t+1} = f_c^x((\Sigma_{il}^{t+1})^2,R_{il}^{t+1})\ , \tag{15}$$

we initialize $\omega_{\mu l}^{t=0} = y_{\mu l}$, $a_{il}^{t=0}$ and $v_{il}^{t=0}$ as the mean and variance of the assumed distribution $P_X(\cdot)$, and iterate these equations until convergence. At every time-step the quantity $a_{il}$ is the estimate for the signal element $x_{il}$, and $v_{il}$ is the approximate error of this estimate. The estimate and its error for the distortion parameter $d_\mu$ can be computed as

$$k_\mu^{t+1} = \left.\frac{\partial}{\partial\theta}G(y_{\mu\cdot}^{t+1},\omega_{\mu\cdot}^{t+1},V_{\mu\cdot}^{t+1},\theta)\right|_{\theta=0} \quad\text{and}\quad l_\mu^{t+1} = \left.\frac{\partial^2}{\partial\theta^2}G(y_{\mu\cdot}^{t+1},\omega_{\mu\cdot}^{t+1},V_{\mu\cdot}^{t+1},\theta)\right|_{\theta=0}. \tag{16}$$

By setting $P_D(d_\mu) = \delta(d_\mu - d_\mu^{\text{true}})$, and simplifying eq. (8), readers familiar with the work of Rangan [14] will recognize the GAMP algorithm in eqs. (12-15). Note that for a general transfer function $h$ the generating function $G$ (8) has to be evaluated numerically. The overall complexity of the Cal-AMP algorithm scales as $O(MNP)$ and hence shares the scalability advantages of AMP [12].

## 2.1 Cal-AMP for the product transfer function

In the numerical section of this paper we will focus on a specific case of the transfer function $h(z_{\mu l},d_\mu,w_{\mu l})$, defined in eq. (2). We consider the measurement noise $w_{\mu l}$ to be Gaussian of zero mean and variance $\Delta$. This transfer function was considered in the work of [9] and we will hence be able to compare the performance of Cal-AMP directly to the convex optimization investigated in [9]. For the product transfer function eq. (2) most integrals requiring a numerical computation in the general case are expressed analytically and we can replace equations (13) by:

$$e_{\mu l}^{t+1} = \frac{k_\mu^t y_{\mu l} - \omega_{\mu l}^t}{V_{\mu l}^t + \Delta}\ , \qquad h_{\mu l}^{t+1} = \frac{1}{V_{\mu l}^{t+1} + \Delta} - \frac{l_\mu^t y_{\mu l}^2}{(V_{\mu l}^{t+1} + \Delta)^2}\ , \tag{17}$$

$$(C_\mu^{t+1})^2 = \left[\sum_n \frac{y_{\mu n}^2}{V_{\mu n}^{t+1} + \Delta}\right]^{-1}\ , \qquad T_\mu^{t+1} = (C_\mu^{t+1})^2 \sum_n \frac{y_{\mu n}\omega_{\mu n}^{t+1}}{V_{\mu n}^{t+1} + \Delta}\ , \tag{18}$$

$$k_\mu^{t+1} = f_a^d((C_\mu^{t+1})^2,T_\mu^{t+1})\ , \qquad l_\mu^{t+1} = f_c^d((C_\mu^{t+1})^2,T_\mu^{t+1})\ . \tag{19}$$

where we have introduced the functions $f_a^d$ and $f_c^d$ similarly to those in eq. (10), except the expectation is made w.r.t. to the measure

$$\mathcal{M}_D(d,C^2,T) = \frac{1}{Z(C^2,T)}P_D(d)|d|^P\ e^{-\frac{(d-T)^2}{2C^2}}\ . \tag{20}$$

## 3 Experimental results

Our simulations were performed using a MATLAB implementation of the Cal-AMP algorithm presented in the previous section, that is available online [22]. We focused on the noiseless case $\Delta = 0$ for which exact reconstruction is conceivable. We tested the algorithm on randomly generated Gauss-Bernoulli signals with density of non-zero elements $\rho$, normally distributed around zero with

unit variance. For the present experiments the algorithm is using this information via a matching distribution $P_X(x_{il})$. The situation when $P_X$ mismatches the true signal distribution was discussed for AMP for compressed sensing in [21].

The distortion parameters $d_\mu$ were generated from a uniform distribution centered at $d = 1$ with variance $\sigma^2$ and width $2\sqrt{3}\sigma$. This ensures that, as $\sigma^2 \to 0$, the results of standard compressed sensing are recovered, while the distortions are growing with $\sigma^2$. For numerical stability purposes, the parameter $\sigma^2$ used in the update functions of Cal-AMP was taken to be slightly larger than the variance used to create the actual distortion parameters. For the same reasons, we have also added a small noise $\Delta = 10^{-17}$ and used damping in the iterations in order to avoid oscillatory behavior. In this noiseless case we iterate the Cal-AMP equations until the following quantity $\mathrm{crit} = \frac{1}{MP} \sum_{\mu l} \left( k_\mu y_{\mu l} - \sum_i F_{\mu i} a_{il} \right)^2$ becomes smaller than the numerical precision of implementation, around $10^{-16}$, or until that quantity does not decrease any more over 100 iterations.

Success or failure of the reconstruction is usually determined by looking at the mean squared error (MSE) between the true signal $\mathbf{x_l^0}$ and the reconstructed one $\mathbf{a_l}$. In the noiseless setting the product transfer function $h$ leads to a scaling invariance and therefore a better measure of success is the cross-correlation between real and recovered signal (used in [10]) or a corrected version of the MSE, defined by:

$$\mathrm{MSE}^{\mathrm{corr}} = \frac{1}{NP} \sum_{il} \left( x_{il}^0 - \hat{s} a_{il} \right)^2 , \quad \text{where} \quad \hat{s} = \frac{1}{M} \sum_\mu \frac{d_\mu^0}{k_\mu} \tag{21}$$

is an estimation of the scaling factor $s$. Slight deviations between empirical and theoretical means due to the finite size of $M$ and $N$ lead to important differences between MSE and $\mathrm{MSE}^{\mathrm{corr}}$, only the latter truly going to zero for finite $N$ and $M$.

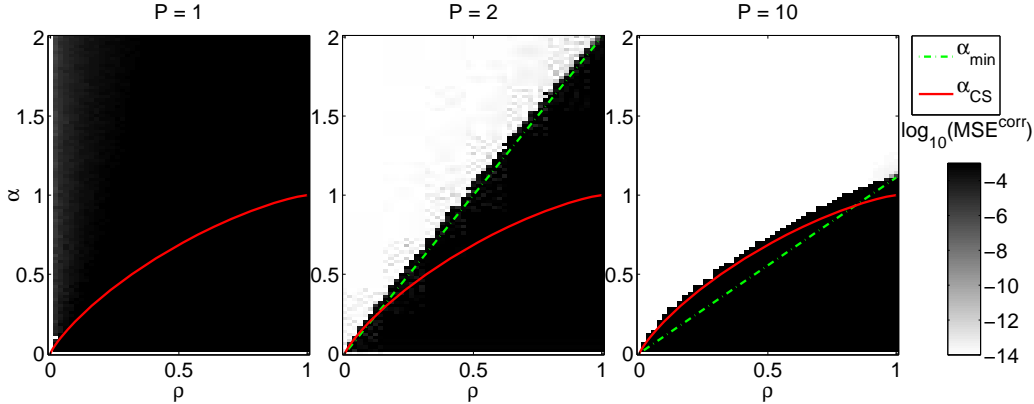

Figure 2: Phase diagrams for different numbers $P$ of calibrating signals: The measurement rate $\alpha = M/N$ is plotted against the density of the signal $\rho = K/N$. The plotted value is the decimal logarithm of $\mathrm{MSE}^{\mathrm{corr}}$ (21) achieved for one random instance. Black indicates failure of the reconstruction, while white represents perfect reconstruction (i.e. a MSE of the order of the numerical precision). In this figure the distortion variance is $\sigma^2 = 0.01$ and $N = 1000$. While for $P = 1$ reconstruction is never possible, for $P > 1$, there is a phase transition very close to the lower bound defined by $\alpha_{\min}$ in equation (3) or to the phase transition line of the pure compressed sensing problem $\alpha_{\mathrm{CS}}$. Note, however, that in the large $N$ limit we expect the calibration phase transition to be strictly larger than both the $\alpha_{\min}$ and $\alpha_{\mathrm{CS}}$. Note also that while this diagram is usually plotted only for $\alpha \leq 1$ for compressed sensing, the part $\alpha > 1$ displays pertinent information in blind calibration.

Fig. 2 shows the empirical phase diagrams in the $\alpha$-$\rho$ plane we obtained from the Cal-AMP algorithm for different number of signals $P$. For $P = 1$ the reconstruction is never exact, and effectively this case corresponds to reconstruction without any attempt to calibrate. For any $P > 1$, there is a sharp phase transition taking place with a jump in $\mathrm{MSE}^{\mathrm{corr}}$ of ten orders of magnitude. As $P$ increases, the phase of exact reconstruction gets bigger and tends to the one observed in Bayesian compressed sensing [15]. Remarkably, for small values of the density $\rho$, the position of the Cal-AMP phase transition is very close to the CS one already for $P = 2$ and Cal-AMP performs almost as well as in the total absence of distortion.

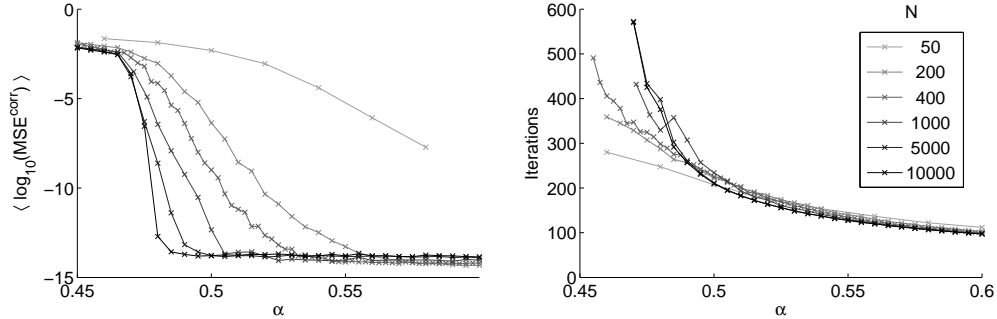

Figure 3: Left: Cal-AMP phase transition as the system size $N$ grows. The curves are obtained by averaging $\log_{10}(\text{MSE}^{\text{corr}})$ over 100 samples, reflecting the probability of correct reconstruction in the region close to the phase transition, where it is not guaranteed. Parameters are: $\rho = 0.2$, $P = 2$, $\sigma^2 = 0.0251$. For higher values of $N$, the phase transition becomes sharper. Right: Mean number of iterations necessary for reconstruction, when the true signal *is* successfully recovered. Far from the phase transition, increasing $N$ does not increase visibly the number of iterations for these system sizes, showing that our algorithm works in linear time. The number of needed iterations increases drastically as one approaches the phase transition.

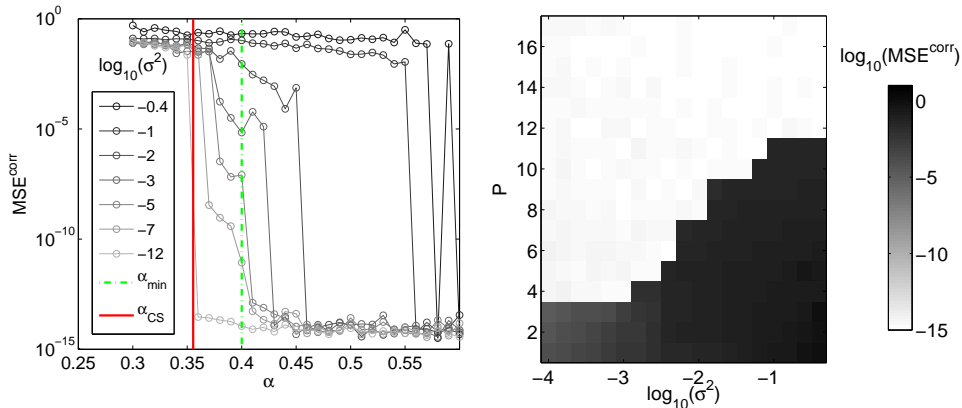

Figure 4: Left: Position of the phase transition in $\alpha$ for different distortion variances $\sigma^2$. The left vertical line represents the position of the CS phase transition, the right one is the counting bound eq. (3). With growing distortion, larger measurement rates become necessary for perfect calibration and reconstruction. Intermediary values of $\text{MSE}^{\text{corr}}$ are obtained in a region where perfect calibration is not possible, but distortions are small enough for the uncalibrated AMP to make only small mistakes. The parameters here are $P = 2$ and $\rho = 0.2$. Right: Phase diagram as the variance of the distortions $\sigma^2$ and the number of signals $P$ vary, for $\rho = 0.5$, $\alpha = 0.75$ and $N = 1000$.

Fig. 3 shows the behavior near the phase transition, giving insights about the influence of the system size and the number of iterations needed for precise calibration and reconstruction. In Fig. 4, we show the jump in the MSE on a single instance as the measurement rate $\alpha$ decreases. The right part is the phase diagram in the $\sigma^2$-$P$ plane.

In [9, 10], a calibration algorithm using $\ell_1$-minimization has been proposed. While in that case, no assumption on the distribution of the signals and of the the gains is needed, for most practical cases it is expected to be less performant than the Cal-AMP if these distributions are known or reasonably approximated. We implemented the algorithm of [9] with MATLAB using the CVX package [23]. Due to longer running times, experiments were made using a smaller system size $N = 100$. We also remind at this point that whereas the Cal-AMP algorithm works for a generic transfer function (1), the $\ell_1$-minimization based calibration is restricted to the transfer functions considered by [9, 10]. Fig. 5 shows a comparison of the performances of the two algorithms in the $\alpha$-$\rho$ phase diagrams. The Cal-AMP clearly outperforms the $\ell_1$-minimization in the sense that the region in which calibration is possible is much larger.

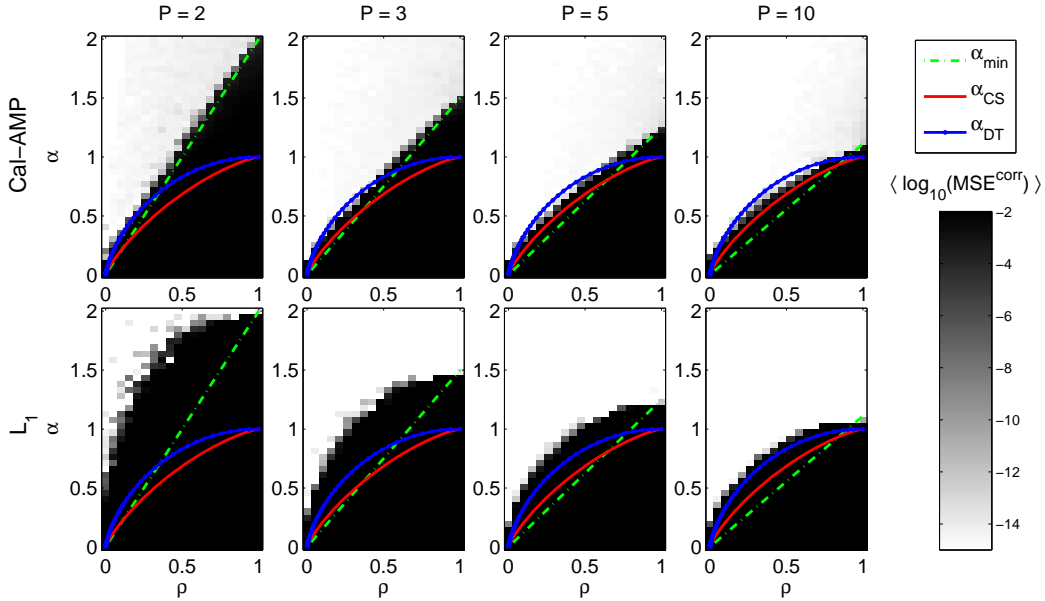

Figure 5: Comparison of the empirical phase diagrams obtained with the Cal-AMP algorithm proposed here (top) and the $\ell_1$-minimization calibration algorithm of [9] (bottom) averaged over several random samples; black indicates failure, white indicates success. The area where reconstruction is possible is consistently much larger for Cal-AMP than for $\ell_1$-minimization-based calibration. The plotted lines are the phase transitions for CS without unknown distortions with the AMP algorithm ($\alpha_{CS}$, in red, from [21]), and with $\ell_1$-minimization (the Donoho-Tanner transition $\alpha_{DT}$, in blue, from [11]). The line $\alpha_{\min}$ is the lower counting bound from eq. (3). The advantage of Cal-AMP over $\ell_1$-minimization calibration is clear. Note that in both cases, the region close to the transition is blurred due to finite system size, hence a region of grey pixels (again, the effect is more pronounced for the $\ell_1$ algorithm).

## 4    Conclusion

We have presented the Cal-AMP algorithm for blind calibration in compressed sensing, a problem where the outputs of the measurements are distorted by some *unknown* gains on the sensors, eq. (1). The Cal-AMP algorithm allows to jointly infer sparse signals and the distortion parameters of each sensor even with a very small number of signals and is computationally as efficient as the GAMP algorithm for compressed sensing [14]. Another advantage w.r.t. previous works is that the Cal-AMP algorithm works for generic transfer function between the measurements and the readings from the sensor, not only those that permit a convex formulation of the inference problem as in [9, 10]. In the numerical analysis, we focussed on the case of the product transfer function (2) studied in [9]. Our results show that, for the chosen parameters, calibration is possible with a very small number of different sparse signals $P$ (i.e. $P = 2$ or $P = 3$), even very close to the absolute minimum of measurements required by a counting bound (3). Comparison with the $\ell_1$-minimizing calibration algorithm clearly shows lower requirements on the measurement rate $\alpha$ and on the number of signals $P$ for Cal-AMP. The Cal-AMP algorithm for blind calibration is scalable and simple to implement. Its efficiency shows that supervised training is unnecessary. We expect Cal-AMP to become useful in practical compressed sensing implementations.

Asymptotic analysis of AMP can be done using the state evolution approach [12]. In the case of Cal-AMP, however, analysis of the resulting state evolution equations is more difficult and has hence been postponed to future work. Future work also includes the study of the robustness to the mismatch between assumed and true distribution of signal elements and distortion parameters, as well as the expectation-maximization based learning of the various parameters. Finally, the use of spatially coupled measurement matrices [15, 16] could further improve the performance of the algorithm and make the phase transition asymptotically coincide with the information-theoretical counting bound (3).

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
