[Reviews · NeurIPS 2013]

Submitted by Assigned_Reviewer_2

The authors considered blind (i.e., unsupervised) calibration of hardware implementation of compressive sensing. They extended the approximate message passing algorithm (AMP) used in compressive sensing to the case of blind calibration, and hence proposed the calibration-AMP algorithm. The authors performed numerical simulation on synthetic data to demonstrate the superiority of the proposed calibration-AMP algorithm.

My main concern of the paper is that the advantage of the proposed method is not convinicingly demonstrated. No theoretic results are offered (although admitedly this is often difficult for Bayesian methods), and simulations results are restricted to synthetic data.

Secondly, as the author menthioned, the derivation of the algorithm largely follows previous literature [12,13,14,15], and hence the contribution is mostly adapting previous methods to a new problem, and appears to be incremental.

It also appears, in my opinion, that the paper is not a typical machine learning paper (e.g., no paper from major machine learning conferences/journals appears in the reference list). But this is a minor point.
Summary: My main concern of the paper is that the advantage of the proposed method is not convinicingly demonstrated. No theoretic results are offered (although admitedly this is often difficult for Bayesian methods), and simulations results are restricted to synthetic data.

I have taken into accout the author feedback.

Submitted by Assigned_Reviewer_5

The paper considers an extension of the standard compressed sensing (CS) problem. Whereas usually we are given y=Ax and wish to recover x, here we assume that y=Ax+noise(theta) and we wish to recover both x and theta (i.e. the noise parameters). A graphical model s constructed (figure 1) and BP is run on the graphical model. Similar to the recently proposed AMP approach to CS, the BP equations can be shown to simplify into Gaussian message passing. The main result of the paper is that empirically, the BP approach works very well (close to the theoretical limit and better than L1 minimization).

I think this paper could be interesting to people who work on analyzing BP (of which there are quite a few in the nips community) since it shows a nice empirical result on a very dense graphs with many loops. Of course given the success of AMP for the standard CS problem, this empirical result is not all that surprising, but I am happy to see people exploring the extent to which the AMP success is limited to CS.

The main drawback of the paper is that given the AMP results for CS, the novelty here is rather limited. This is a small extension of the approach that was used for the standard CS problem.

Another weakness is that unlike the AMP results for CS, here all the phase transitions are calculated empirically. In other words, there is no proof that BP will solve the problem correctly for the "white" part of the phase diagram. In the standard AMP there are also "state evolution" equations that can be used to prove the phase diagram but the authors defer this derivation to future work.
Summary: Interesting empirical success of BP on a very dense graph with many loops. Limited novelty given the AMP results for CS and no proof of correctness.

Submitted by Assigned_Reviewer_6

The authors provide a nice reconstruction algorithm for compressed sensing in the presence of distortions. It is a minor generalization of previous work but appears more useful in applications that are sometimes encountered.
QUALITY: The authors address an important problem and provide a good solution. The algorithmic development seems sound, though I did not check the details.
CLARITY: The writing is quite clear, except for the omitted derivation of their algorithm. However, they are quite pressed for space, and I think they have made a reasonable compromise. I would like to see further explanation but I will investigate some of the references they provide. Nonetheless, several more explanatory lines would be warranted for non-experts, and could readily be fit in by using more two-column formatting for their equations and scaling the vertical axis of a figure or two.
ORIGINALITY: This work appears to be incremental progress on using message passing algorithms for compressed sensing, but it seems to be a useful compromise of speed and generality (general distortions, but less uncertainty than in dictionary learning).
SIGNIFICANCE: It seems likely that some researchers will use their algorithm in solving practical problems.

Lines 150-151 have an error, because the marginal distribution over the signal components and the distortion parameters are identical.
Lines 181-182 have been too abbreviated. The assumptions and consequences should be spelled out more.
Line 228: The running time may be comparable to GAMP but I don't know what that running time is. The relevant goodness should be better explained.

I would like to see some evidence that their algorithm is successful with non-product distortions, as they only compare their performance to that of existing algorithms on particular convex problems.
Summary: The authors provide a nice reconstruction algorithm for compressed sensing in the presence of distortions. It is a minor generalization of previous work but appears more useful in applications that are sometimes encountered.

Submitted by Assigned_Reviewer_7

The "blind calibration" problem involves a compressed sensing problem where, when observing a M-dimensional measurement of N-dimensional sparse signals x_1,x_2,..., each of the M measurement sensors is distorted in some parametrized way, where the parameter can vary over the sensors. Therefore the goal is to be able to recover these distortion parameters d_1,...,d_M at the same time as estimating the sparse signals.

This paper proposes an effective message-passing algorithm "C-AMP" to solve the two problems simultaneously. The authors use a simple dimensionality argument to derive a lower bound alpha_min on the sampling ratio alpha=(measurement dimension M)/(signal dimension N), which is needed for recovery to be possible. In the simulation results, C-AMP performs very well, with successful signal recovery nearly everywhere above the threshold max{alpha_CS,alpha_min} where alpha_CS is the phase transition for ordinary compressed sensing (i.e. the setting where there are no unknown calibration parameters). In contrast, L1 minimization (which does not take the calibration issue into account) does not perform nearly as well.

I did find the initial presentation of the problem to be somewhat confusing. Because blind calibration is presented after background is given on supervised calibration, it gives the impression that the goal is again the estimation of the distortion parameters d_mu, rather than signal recovery (the sparse x_l's). It would perhaps be clearer to say something like, "Given a sparse signal recovery problem, if we were not able to previously estimate the distortion parameters via supervised calibration, we will need to estimate the unknown signals and the unknown distortion parameters simultaneously - this is known as blind calibration."








Other comments...

-- 062: z_{\mu l} should be z_{\mu}?

-- 063-064: not clear if w is the noise or w is noise+z? I assume that w is the noise Delta; might be clearer to say "one usually considers an iid Gaussian noise with variance Delta, which will then be added to z". Also there is an extra "one" in this sentence.

-- 101-102: this is hard to follow. Perhaps specify what is rho.

-- 108: "the use if AMP" - > "the use of AMP"

-- 132: very nice intuitive derivation of the lower bound

-- equation in line 146 is very clear and easy to read; in-line equations in lines 178-180 are harder to read & understand

-- 149: "Z" is not the best notation for the normalizing constant since "z" is used for something unrelated

-- 158-159: "NPM messages" and "PM messages" should maybe be "pairs of messages" since ther are m's and \tilde{m}'s?

-- 263-264: This line is a bit awkward to read, perhaps rewrite

-- 268-269: not sure what it means to have a uniform distribution with a mean & variance parameter - I assume it means that sigma^2 determines the width of the interval, i.e. the width is chosen such that variance is equal to sigma^2? This is an unusual parametrization and it might be easier to just give the width itself.

-- 300: what is the green line in the central figure?

-- 380: these figures & results are very clear. It might be nice to discuss alpha_min a bit more - give some intuition for how alpha_min is not a phase transition, since alpha_CS sometimes lies above alpha_min in these figures; perhaps the maximum of alpha_min and alpha_CS functions as a phase transition? Or is there some single bound that might unite the two?

-- I think it's more standard, in phase transition diagrams, to use light regions to indicate successful signal recovery and dark regions when the signal cannot be recovered.
Summary: The paper gives an effective message-passing algorithm to address the blind calibration problem, where we would like to recover sparse signals from lower-dimensional measurements, but these measurements themselves need to be calibrated. The algorithm is very effective in simulations and fits closely to a lower bound that is derived with a simple dimensionality argument.

Submitted by Assigned_Reviewer_8

The paper employs the the approximate message passing framework recently developed in compressed sensing for the blind calibration problem. They then use extensive simulations to characterize the phase transition of their algorithm in the noiseless settings. This is a nice paper. However, the presentation can be improved a lot. Here are some suggestions:

(i) The authors are employing the Bayesian setting where they assume both the signal distribution and distortion distribution d_\mu are known. However, they don't mention it in Section 1.1. They also never explicitly mention this assumption and one should realize it from (4). IT will be helpful if they clearly state their assumptions.

(ii) They never provide any details on the ensemble of the measurement matrices they are using. Are they spatially coupled? Are they iid subGaussian? Again stating this clearly in Section 1.1 will be very helpful.

(iii) Even though the authors cite several papers that have considered the same calibration problem, it is still helpful to explain one concrete application and then mention why the assumptions they are making are correct? For instance it is not clear for the reviewer why d_\mu is independent of $\ell$. It seems that the distortion should depend on the measurement and hence on $\ell$.

(iv) The name C-AMP seems to be used in other message passing papers, for complex-valued signals. Other names might make the concept more clear.



()
Summary: Nice paper that requires a revision for clarification purposes.
Author Feedback

Author rebuttal: Dear NIPS committee,

We thank the reviewers for their valuable feedback, and their overall appreciation of our work. We will make sure to clarify the ambiguities they raised in the final version of our paper. Below we wish to comment on the part of the points made that we consider contestable.

First, we would like to make a general comment about the "Impact Score", 3 reviewers granted us with score 1, and 2 with score 2. We would definitely agree that our work is different from a typical NIPS submission. Its strong side is that (based on non-rigorous heuristic arguments) we were able to design a very efficient message-passing based algorithm for a hard inference problem that resists to more traditional approaches. The weaker side, as basically all the reviewers pointed out, is that our paper does not include any theorem proving the performance of the algorithm. Indeed, establishing such theorems would be a very challenging task. We would be very grateful if the NIPS committee gave us the opportunity to present this work and this general direction of research at the conference.

Here is now a more specific answer:

Reviewer_2
---------

Reviewer: "Secondly, as the author mentioned, the derivation of the algorithm largely follows previous literature [12,13,14,15], and hence the contribution is mostly adapting previous methods to a new problem, and appears to be incremental."

Answer: We respectfully disagree. The derivation of the algorithm is not a trivial generalization of the AMP one and our updates for the estimates of the distortion parameters are nor intuitive neither easy to guess. The corresponding graphical model being more complex than the one corresponding to compressed sensing, the derivation is instead quite involved. Since we did not have the place to provide the full derivation and gave instead the references [12,13,14,15] (where AMP for compressed sensing was derived), we perhaps gave the (wrong) impression that the generalization was simply an incremental one. This is not the case. Our approach allowed us to solve efficiently an inference problem that cannot be solved by a simple generalization of AMP (as far as we know) and that is firmly resisting other more traditional ones (convex relaxations etc). Hence it is not true that we barely adapt an "existing method".

Reviewer_5
---------

We believe this reviewer misunderstood the definition of our problem. The blind calibration problem is more complicated than estimating theta and x from y=Ax+noise(theta), which would be a simple generalization of compressed sensing. We believe that his claim that "This is a small extension of the approach that was used for the standard CS problem." stems from this misunderstanding.

He correctly states that our validation of the method is empirical. We have derived the corresponding "state evolution" already, but since blind calibration is a non-trivial generalization of the compressed sensing problem, the corresponding equations are complicated and it is not clear to us at this point if they are amenable to rigorous analysis such as the one done for "state evolution" for AMP for compressed sensing. Though empirical, we believe that Figure 3 gives convincing evidence that our algorithm succeeds with high probability above the phase transition.

Reviewer_ 6
---------

We thank this referee for his supportive comments.

"It is a minor generalization of previous work ...." we disagree with this statement see our answer to Reviewer_2.

Lines 150-151, yes there is a misprint that we will correct. Lines 181-182 will be expanded. Line 228 the running time is O(MN), we will add more quantitative values.

Reviewer_ 7
---------

We thank this referee for his supportive comments and for pointing several places where our paper is unclear, this is immensely valuable to us and we will adjust the manuscript accordingly.
We would like to clarify one point though: the L1 minimization algorithm we use as a comparison (used by Gribonval, Chardon and Daudet in [9]), does take into account calibration (else it could never achieve perfect recovery), but it is still not as performant as the algorithm we propose.

Reviewer_ 8
---------

We also thank this referee for supportive comments and for pointing several places where our paper is unclear, in detail:

(i) We will state more clearly our Bayesian-like assumptions.
(ii) We work with iid gaussian measurement matrices (indeed we expect that using the spatially coupled ones would further improve the performance).
(iii) The different measurements are indexed by $\mu$. In applications, each $\mu$ can correspond to one sensor. And the parameter $d_\mu$ is related to the mis-calibration of that sensor. The index $\ell$ stands for different signal samples that are used for the blind (unsupervised) calibration, the mis-calibration parameters do not depend on the sample index, since the measurement hardware is independent of the realization of the sample. We will give an example of a specific application in the revised version.
(iv) We realized that the abbreviation C-AMP was used for complex-AMP, we will hence revise it to Cal-AMP.